# Enhanced Detection Systems of Filling Rates Using Carbon Nanotube Cement Grout

**DOI:** 10.3390/nano10010010

**Published:** 2019-12-18

**Authors:** Heeyoung Lee, Seonghoon Park, Sanggyu Park, Wonseok Chung

**Affiliations:** 1Department of Civil Engineering, Chosun University, Gwangju 61452, Korea; 2Department of Civil Engineering, Kyung Hee University, Yongin-si, Gyeonggi-do 17104, Korea; 9254sunghoon@gmail.com (S.P.); gyu2644@gmail.com (S.P.)

**Keywords:** carbon nanotube, grout, void detection, thermal imaging analysis, electrical resistance, field emission scanning electron microscopy (FE-SEM)

## Abstract

The addition of small amounts of carbon nanotubes (CNTs) to cement-based materials modifies their thermal and electrical characteristics. This study investigated the void detection and filling rates of cement grout with multi-walled carbon nanotubes (MWCNTs). MWCNT grouts of 40 mm × 40 mm × 160 mm were fabricated. Specimens were tested by thermal imaging, electrical resistance analyses, and magnetic field tests. The experimental parameters were the concentration of MWCNT and the grout filling rate. The filling rate was investigated by measuring resistance and magnetic field changes with respect to cross-sectional area, taking the voids into consideration. The results of the thermal image tests indicate that 1.0 wt % MWCNT cement grout is optimal for void detection.

## 1. Introduction

A number of recent studies have focused on transforming cement-based construction materials by mixing them with nanomaterials to produce functional construction materials [1,2,3,4,5,6,7,8,9,10]. The resulting materials exhibit improved mechanical characteristics such as compressive strength and bond strength, and enhanced electrical and heat conduction [1,2,3,4,5,6,7,8,9,10,11,12,13,14,15,16,17,18,19,20].

One nanomaterial that has attracted considerable attention is the carbon nanotube (CNT), which is a carbon allotrope. CNTs are used as multifunctional materials due to their thermal and electrical properties [13,14,15]. Therefore, many researchers in the fields of composite materials and polymer materials have attempted to improve and functionalize materials using CNTs. However, compared to other materials, few studies have examined the mixing of CNTs with cement materials. This is because cement materials are less homogeneous than other materials, making it difficult to produce homogeneous CNT mixtures with good properties. Therefore, further research on mixtures of cement-based materials is needed. If CNTs and cement-based materials can be mixed properly, the resulting material may have increased mechanical strength and electrical and thermal properties that can be adjusted from non-conductive to conductive [11,12,16,17,18,19,20]. Furthermore, these materials could be used for applications such as curing, melting snow, deicing via heat conduction, and measuring changes in electrical resistance via electrical conduction.

As part of the expansion and improvement of infrastructure around the world, prestressed concrete (PC) bridges have been widely constructed. In PC bridges, grouting is implemented by densely filling ducts with cement-based materials to shield prestressed steel strands from external damaging substances such as chloride. If poor grout charging occurs from the bleeding of water and shrinkage, deicing chemicals and moisture may penetrate the grout and cause the strands to corrode and fracture. This poor grout charging significantly affects the structural safety of PC bridges. If methods for measuring and assessing the filling rate of PC bridges are available, such problems could be detected in advance through periodic monitoring. Therefore, the aims of this study were to investigate grout using nanomaterials and to use the electrical and thermal conductivity of these nanomaterials for grout void detection inside ducts.

Experimental studies have examined mixtures of nanomaterials with cement-based materials [1,2,3,4,5,6,7,8,9,10]. Researchers have investigated the excellent electrical and thermal conductivity afforded by CNTs to realize various functions based on heating characteristics [11,12,13,14,15,16,17,18,19,20]. Lee et al. [21] fabricated cement mortar incorporating single-walled carbon nanotubes (SWCNTs) and multi-walled carbon nanotubes (MWCNTs); the authors experimentally studied the heating characteristics of different CNT types, concentrations, and mixing methods. The SWCNT cement mortar showed better heating characteristics than the MWCNT mix; however, as the CNT concentration increased, agglomeration occurred, and the heating characteristics degraded. The heating properties of the CNT cement based on mortar were meaningfully affected by its dispersibility, which is CNT type, concentration, and incorporation method. In this study, the filling rate in grout was measured using SWCNTs and MWCNTs [22].

Kim et al. [23] investigated the heat-induced mechanical characteristics of cement composites mixed with 0.1–2.0 wt % CNTs under supply voltages of 3–20 V. In the heating experiment, the temperature of the 2.0 wt % CNT specimen increased by ~70 °C within 30 min. The heating capability decreased when the experiment was repeated, and the decrease for the 1.0 wt % and 2.0 wt % specimens was more pronounced than that for 0.3 wt % and 0.6 wt % samples. Gomis et al. [24] fabricated plate-shaped specimens with dimensions of 100 mm × 100 mm × 10 mm using graphite powder, carbon fiber powder, carbon fiber, carbon nanofiber (CNF), and CNTs. For a specimen with 1.0 wt % CNTs, ~110 g of ice was thawed within 25 min under a supply voltage of 90 V.

Zhang and Li [25] studied snow-melting and deicing systems in MWCNT cement-based paved roads that used a CNF polymer as a heat source. For an MWCNT concentration of 3.0 wt %, the thermal conductivity was measured to be 2.83 W/(m·K). The snow-melting and deicing time increased linearly with ice thickness and decreased linearly with an increase in the surrounding temperature. Zhao et al. [26] examined the deicing capability of carbon fiber heating wires (CFHWs) embedded in a concrete slab. Four CFHWs were embedded at 100-mm intervals, and a heating experiment was conducted in a refrigerator at −25 °C with a heating intensity of 1134 W/m^2^. The surface temperature of the slab increased at a rate of approximately 0.17 °C/min and exceeded 0 °C after 2.5 h. Chung [27] examined self-heating structural materials that are useful for deicing and space heating, such as carbon fiber mats, carbon fiber, and graphite. The heating characteristics were determined by mixing electrically conductive fibers with cement and increasing the heat resistance. The experimental results showed that the cement polymer constructed with a carbon fiber mat had the best self-heating performance.

In addition to mechanical strength and heating characteristics, studies have also analyzed the electrical properties of cement-based mortar with CNTs. Konsta-Gdoutos and Aza [28] examined the electrical resistance of composites mixed with 0.1 wt % CNT and 0.3 wt % CNF. The resistance changes under repeated compressive loads were measured using a four-pole method through electrodes embedded in the specimens. The resistance of the 0.1 wt % CNT specimens was lower than that of the 0.3 wt % CNF specimen and showed a larger range of resistance change under repeated loads. Han et al. [29] investigated the effects of repeated loads on MWCNT cements with various moisture concentrations and tested the piezoresistivity. The study concluded that the piezoresistivity of the MWCNT cements is strongly related to moisture concentration. The effects of CNT concentration and water-to-cement ratio on piezoresistivity sensitivity were later investigated by comparing the electrical resistance under compressive loads [30]. The study confirmed that the piezoresistivity was most sensitive at a CNT concentration of 0.1 wt % and increased with higher water-to-cement ratio. The load applied to a specimen could be determined by experimentally measuring the electrical resistance of the CNT cement composite.

Various experimental studies on the structural stability of grout have focused on the corrosion and filling rates. Hong et al. [31] assessed the integrity of grouting in umbrella arch methods with ultrasonic waves. These experiments applied the hammer impact reflection method, and a guided ultrasonic wave was used to measure the propagation velocities of specimens with longitudinal grouted ratios of 25%, 50%, 75%, and 100%. The results indicate the difficulty of assessing the grouted ratio using the change in guided ultrasonic wave velocity. Liu et al. [32] utilized travel-time tomography to evaluate the duct filling rates of PC structures. Visualization of the tomography reconstruction technique became more effective with an increase in the number of sensors. Conventional approaches such as x-radar, radar, supersonic, and impact echo methods are individually inaccurate for such evaluations. Thus, to evaluate the grout filling rate in ducts, Mori et al. [33] tried to develop the impact echo method. They experimentally showed that grout voids inside ducts could be detected by analyzing the relationship between surface velocity and standard signal.

This study aims to fabricate MWCNT cement grout by admixing MWCNTs and to carry out void detection by conducting thermal imaging analysis, electrical resistance change analysis, and magnetic field strength analysis. Electrical resistance change analysis was used to detect the filling rate by measuring the resistance change on the cross section. The electrical characteristics and magnetic field analysis of the MWCNT cement grout varied according to the MWCNT concentration. Thus, the concentrations of MWCNTs used in this study were relatively low and high—namely, 0.1 wt % and 1.0 wt % (with respect to the cement weight). The void detection was studied by comparing the thermal effects and electrical properties of MWCNT grout composites.

## 2. Experimental Procedures

### 2.1. Characteristics of Multi-Walled Carbon Nanotubes

Naturally occurring allotropes of carbon include CNTs, graphite, and graphene. CNTs in the form of MWCNTs were discovered when carbon spin-offs attached to graphite electrodes were monitored by microscopy [12,13]. Under equivalent conditions, CNTs are more than 100 times stronger than steel. Their electrical characteristics result from the unique electrical structure and linear one-dimensional structure of graphite; CNTs have low electrical resistance.

Table 1 shows the physical properties of individual MWCNTs [21,22]. The average diameter of the MWCNTs was 9.5 nm and the average length was 1.5 μm. The MWCNT samples were obtained from Dittotechnology Co. Ltd. (Gunpo city, Republic of Korea), and class 1 ordinary Portland cement was used for the cement. MWCNTs exhibit better mechanical and physical characteristics than other nanomaterials; in addition, they can be produced easily and inexpensively. CNTs also have excellent dispersibility, making them suitable for mixing with cement-based materials. This study used MWCNTs, which have inferior physical properties to SWCNTs, but can disperse at higher concentrations. CNT dispersion is a crucial factor for use in the construction field. To ensure that their characteristics are manifested, CNTs must be mixed homogeneously with cement-based materials. However, CNTs exist in bundles because of agglomeration due to van der Waals (vdW) forces. Previous studies have used various methods to reduce vdW forces and achieve homogeneous CNT dispersion. Generally, a chemical surface treatment method is employed to reduce the strong attraction between CNTs; then, a mechanical agitation method is used to disperse them [34]. Studies have found that a combination of surfactant and ultrasonic treatments improved the dispersibility of CNTs [4,11,19,35]. Other studies have investigated CNT dispersion methods to derive the optimal mixing technique for cement-based materials [36,37,38].

### 2.2. Fabrication of Specimens

The test parameters in this study were the grout filling rate and CNT concentration. To assess the filling rate, the temperature changes and electrical resistance changes of MWCNT grouts were investigated. Three specimens were fabricated for each parameter. Grout specimens with dimensions of 40 mm × 40 mm × 160 mm were fabricated, simulating PC ducts [39]. Table 2 lists the parameters of the CNT cement ground void detection experiment. Group #1 consisted of ordinary cement grout, which was used for comparison. Group #2 was the MWCNT-mixed grout group, which was divided into specimens with low and high CNT concentrations of 0.1 wt % and 1.0 wt %, respectively (with respect to the cement weight). Figure 1 shows the grout filling rates for voids in the middle of the specimens. In Group #1, filling rates of 50% and 100% were selected. In Group #2, filling rates of 25%, 50%, 75%, and 100% were selected. All specimens were cured for 28 days. The first letters of each specimen name indicate the material used: “OPC” refers to the ordinary cement grout, and “MW” refers to the MWCNT group. In Group #2, “0.1” and “1.0” represent the CNT concentration, and “25,” “50,” “75,” and “100” represent the grout filling rate.

The specimens were fabricated in accordance with grout cement [39,40,41]. The water-to-cement ratio was fixed as 0.5, the sand-to-cement ratio was set to 2.5, and the curing time was 28 days. A rectangular transparent duct with dimensions of 40 mm × 40 mm × 160 mm was used to fabricate each specimen. The used grout was mixed by Grade A in ASTM C11-7 [41,42]. Steel mesh electrodes with dimensions of 50 mm × 70 mm were used for the voltage supply. Thermocouples were set up for temperature measurements.

Figure 2 shows the specimen fabrication process. The cement, sand, and solution were measured in accordance with the mixing ratio for each parameter (Figure 2a). To produce a homogeneously dispersed CNT solution, a surfactant was added to the undispersed CNT solution, and then the solution was ultrasonicated. In this research, MWCNTs were dispersed for 4 h by sonicating at 22 kHz using an ultrasonic processor.

The selection of surfactant concentration and ultrasonic dispersion time for homogenous dispersion is important for cement composites mixed with MWCNTs of 1.0 wt %, which is a high concentration. The surfactant and ultrasonic time were selected through numerous experiments for homogenous dispersion within the MWCNT-cement composite structure. In this study, the MWCNT solution was manufactured using the surfactant sodium dodecyl sulfate (SDS). For the fabrication of the solution, the mass ratio of MWCNT:SDS was chosen as 1:4. For the homogeneous dispersion of a high concentration of MWCNT mixed with the cement, the solution was subjected to the ultrasonic dispersion process. The MWCNT solution was subjected to an ultrasonic wave dispersion process through 30 W of energy for four hours.

The cement and sand were mixed using a mixer with a maximum power of 700 W for 2 min (Figure 2b). The CNT solution was then added and mixed for 3 min using the mixer, as shown in Figure 2c. Figure 2d displays the process of creating a void in advance in the middle of each transparent duct. After mixing, the specimens were compacted. Half of the transparent duct was filled with grout and compacted; steel mesh electrodes were then inserted 10 mm apart at both ends of the specimen, as shown in Figure 2e. Then, the remaining half was filled with grout and compacted; thermocouples were embedded 10 mm apart at both sides of the void (Figure 2f). The completed specimens were cured in a thermohygrostat for 24 h, as shown in Figure 2g, and then subjected to room temperature curing at 23 ± 2 °C (Figure 2h). After curing for 28 days, all specimens were dried completely. The hydration and strength of the MWCNT cements used in this study were similar to those of ordinary cement grouts [43].

### 2.3. Test Methods

Before being subjected to a voltage, the specimens were installed on an insulated rubber plate. The power supply (EX200-1) was linked to the electrodes, as shown in Figure 3a; a voltage of 50 V was applied until no further temperature change occurred. DC voltage was supplied to provide a current with constant magnitude and direction. The temperature change was measured by linking a thermocouple to the datalogger. When the temperature reached the maximum value, thermal imaging analysis was performed using a Testo 882 thermal imager (specifications in Table 3). A four-probe measurement was used to measure the electrical performance of the MWCNT cement grout specimen. The electrical resistance was measured until a constant value was obtained by connecting a digital multimeter (Keithley 2701) to the steel mesh (Figure 3b). A four-probe measurement method was used to minimize the electrical contact resistance between the electrode of the specimen and the measurement probe. After touching the four probes on the specimen to be measured, the current was supplied to the outside probe. The digital multimeter measured the voltage according to the supplied input current value through the probe inside. The resistance of the MWCNT cement grout specimen could be easily obtained through a current–voltage (I–V) graph of a digital multimeter.

Equation (1) shows the Joule heating (Q); *I* is the current; *V* is voltage and *t* is time. *R* is the resistance.
(1)$$Q=V^{2}\cdot t/R$$

A magnetic field is formed around the conductor where the current is generated. The direction of the generated magnetic field is the vertical direction of the current. The magnetic field strength is assessed by applying the Biot–Savart law on a plane. According to Equation (2), the magnetic field strength (*B*) acting at a constant distance (*r*) in the conductor to which the current (*I*) is supplied is proportional to the inverse of the current and distance [44,45,46,47,48]. In this case, (*k*) is a constant and represents the value of $2\cdot 10^{-7}$ T∙m/A.
(2)$$B=k\cdot \frac{I}{r}\;\left(k=2\cdot 10^{-7}T\cdot m/A\right)$$

The MWCNT cement grout specimen has more MWCNT particles in the same cross-section as the filling rate increases. As the filling rate of the grout increases, the electrical conductivity and magnetic field strength of the specimen increase because of the increase in MWCNT particles. The improved electrical conductivity and magnetic field of the specimen showed a high current value at the same voltage. A magnetic field sensor (PASCO) was used to measure the magnetic field of the MWCNT cement grout specimen, as shown in Figure 3c. The distance between the MWCNT cement grout specimen and the magnetic field sensor was fixed at 10 mm. The measurement position of the magnetic field sensor was the center part of the specimen; the supplied voltage was maintained at 50 V. The magnetic field was measured in units of 1 s for 30 min for each specimen. The magnetic field data used the average value of the accumulated magnetic field.

## 3. Experimental Results

For each MWCNT concentration and filling rate, the temperature change and electrical resistance change of the MWCNT cement grout were measured, and thermal imaging analysis was conducted. Three specimens were tested for each parameter, and the average temperature change (ΔT, °C) and electrical resistance (Ω) were measured. Thermal imaging was performed when each specimen attained its maximum temperature, and the imaging results for different filling rates were compared. To examine the filling rate of MWCNT cement grout with different MWCNT concentrations, the internal cross-section of the MWCNT cement grout was snapped using field emission scanning electron microscopy (FE-SEM).

### 3.1. Thermal Characteristics of The MWCNT Cement Grout

Table 4 presents the results of the thermal experiments performed for each filling rate of the MWCNT cement grout. The maximum temperature change was measured in the specimen mixed with 1.0 wt % MWCNT, followed by that with 0.1 wt % MWCNT and then OPC. Figure 4 shows the temperature variations of the OPC specimens at the supply voltage. The temperature of the 100%-filled specimen (OPC-100) increased by 0.65 °C, whereas that of the 50%-filled specimen (OPC-50) increased by 0.85 °C. The temperature increase of OPC-50 was similar to that of OPC-100. Furthermore, the temperature increase in ordinary cement grout was not significant under a 50-V supply. As cement is a nonconductor and does not allow current to flow, no temperature increase was expected. There was no significant difference in the temperature change with filling rate. Figure 5 displays the thermal images of the OPC specimens. It is difficult to identify the shape with the naked eye through a thermal camera because of the outdoor air temperature and increasing temperature of the duct during grouting.

Figure 6a presents the temperature variations of the 0.1 wt % MWCNT specimens. The temperature of the 100%-filled specimen (MW-0.1-100) increased by 0.65 °C, whereas that of MW-0.1-75 increased by 0.65 °C. MW-0.1-50 and MW-0.1-25 underwent temperature increases of 0.75 °C and 0.75 °C, respectively. When a voltage was applied, there was no significant temperature change in any MW-0.1 specimen. Figure 6b shows the temperature variations of the 1.0 wt % MWCNT specimens under a supply voltage of 50 V. Temperature increases of 18.8 °C, 17.3 °C, 12.3 °C, and 9.8 °C were measured in MW-1.0-100, MW-1.0-75, MW-1.0-50, and MW-1.0-25, respectively, indicating 8%, 35%, and 48% lower temperature increases in MW-1.0-75, MW-1.0-50, and MW-1.0-25 than in MW-1.0-100. The temperature increase of the 1.0 wt % MWCNT specimen was noticeably higher than that of the 0.1 wt % MWCNT specimen.

Thermal images of the 0.1 wt % MWCNT specimens are presented in Figure 7. The filling rate in the middle of each specimen could not be clearly confirmed. As the 0.1 wt % MWCNT specimen had a small temperature increase, it was difficult to visually distinguish it from the thermal images. In the MWCNT grout, a temperature increase occurred when the MWCNT and MWCNT networks formed. However, the 0.1 wt % MWCNT is less likely to increase the temperature of the MWCNT specimen because the amount of MWCNT is relatively small. Figure 8 shows thermal images of the 1.0 wt % MWCNT specimens. The filling rate in the middle of each specimen could be clearly verified through thermal imaging. The 100%-filled and 75%-filled specimens exhibited even temperature increases, whereas the 50%-filled and 25%-filled specimens displayed concentrated temperature changes close to the void. The specimen shapes of each filling rate could be confirmed by the naked eye, which suggests the feasibility of thermal imaging analysis based on temperature increase in the internal grout, even during grouting. When the concentration of MWCNT was approximately 0.1 wt % according to the thermal image, the temperature increase was not significant; however, when the MWCNT concentration was greater than 1.0%, a definite temperature change occurred. This tendency is appropriate for the use of a 1.0 wt % concentration of MWCNT for the void detection of grout.

### 3.2. Electrical Characteristics of The MWCNT Cement Grout

Table 5 presents the electrical resistance measurements for each specimen. Figure 9 shows the resistance variations of the OPC specimens according to filling rate. The resistance of OPC-100 was 1390 kΩ, whereas that of OPC-50 was 1440 kΩ. The OPC specimens did not exhibit any obvious change in resistance with filling rate. The resistance variations of the 0.1 wt % MWCNT specimens are shown in Figure 10a. MW-0.1-100 had a measured resistance of 1040 kΩ, compared to 1110 kΩ, 1270 kΩ, and 1390 kΩ for MW-0.1-75, MW-0.1-50, and MW-0.1-25, respectively, which were higher than MW-0.1-100 by 6.7%, 22.1%, and 33.7%. The 0.1 wt % MWCNT specimens showed a much higher resistance close to the OPC specimens, even though they exhibited an obvious resistance change with respect to filling rate. Therefore, void detection using 0.1 wt % MWCNT specimens is not appropriate. Figure 10b displays the resistance variations of the 1.0 wt % MWCNT specimens with respect to the filling rate. MW-1.0-100, MW-1.0-75, MW-1.0-50, and MW-1.0-25 had resistances of 0.449 kΩ, 0.575 kΩ, 0.846 kΩ, and 0.934 kΩ, respectively; these represent 28.1%, 88.4%, and 108.0% increases in resistance for MW-1.0-75, MW-1.0-50, and MW-1.0-25 compared with MW-1.0-100. Thus, the 1.0 wt % MWCNT specimens exhibited an obvious resistance change with respect to filling rate, with resistance increasing as the filling rate decreased. Therefore, the filling rates of the 1.0 wt % MWCNT specimens can be verified through electrical resistance analysis. It should be noted that the electrical resistance of the 1.0 wt % specimens was more than 1000 times less than that of the 0.1 wt % specimens. Void detection was difficult for ordinary cement grout and 0.1 wt % MWCNT cement grout because no electrical conductivity was expected due to relatively higher resistance.

The magnetic field strengths of the OPC specimens are shown in Figure 11. The measured magnetic field of OPC-100 and OPC-50 in Group #1 was 0.0007 × $10^{-5}\cdot T$. As the OPC specimen was made of ordinary grout and sand, the magnetic field did not change with the filling rate. The magnetic strengths of the 0.1 wt % MWCNT specimens are shown in Figure 12a. The magnetic field strengths of MW-0.1-100 and MW-0.1-75 were 0.0010 × $10^{-5}\cdot T$ and 0.0009 × $10^{-5}\cdot T$, respectively, while those of MW-0.1-50 and MW-0.1-25 were 0.0008 × $10^{-5}\cdot T$ and 0.0007 × $10^{-5}\cdot T$. As the filling rate of 0.1 wt % MWCNT cement grout was increased by 25%, the strength of the magnetic field increased by 8.33%. The magnetic field strength of the 0.1 wt % MWCNT cement grout specimen improved compared to that of the OPC. However, the magnetic field strength of the 0.1 wt % MWCNT cement grout was close to that of OPC, similar to the electrical resistance result. It can be concluded that the magnetic field strength of the MW-0.1 wt % specimen is not appropriate when varying the filling rate of the grout.

The magnetic field strengths of the 1.0 wt % MWCNT cement grout specimens are shown in Figure 12b. The magnetic field strengths of MW-1.0-100, MW-1.0-75, MW-1.0-50, and MW-1.0-25 were 2.2271 × $10^{-5}\cdot T$, 1.7391 × $10^{-5}\cdot T$, 1.1820 × $10^{-5}\cdot T$, and 1.0707 × $10^{-5}\cdot T$, respectively. The 1.0 wt % MWCNT cement grout upgraded the magnetic field strength by 17.33% as the filling rate increased by 25%. The 1.0 wt % MWCNT cement grout created a better magnetic field performance than the 0.1 wt % MWCNT or the OPC grout. As the concentration of MWCNT in the grout specimen increased, the amount of current increased in the specimen. The MWCNT cement grout specimens formed a strong magnetic field outside the specimen due to the increased current. Based on this tendency, it can be concluded that MWCNT cement grout can measure the filling rate using electrical characteristics and magnetic field strength.

To examine the network and dispersion inside the CNT cement grout, cross-sections of the specimens were analyzed using FE-SEM. Figure 13 shows the FE-SEM image of the specimen. A thread-shaped MWCNT (red color no. 2) can be observed, which connects the calcium silicate hydrates (C–S–Hs, yellow color no. 1). The OPC specimen in Figure 13a shows only the C–S–H formed by cement hydration. The 0.1 wt % MWCNT specimen in Figure 13b shows the thread-shaped MWCNT, but the number of connections to the cement hydrates was small. Therefore, it is analyzed that the 0.1 wt % MWCNT specimen represents a high resistance. It was apparent that MWCNTs connect the C–S–Hs. The MWCNTs form networks between the cement hydrates; these appear to afford electrical and heat conductivity to the grout. This specimen showed obvious heating effects because the CNTs were evenly distributed and formed a network.

As a result, the 1.0 wt % MWCNT specimen clearly had better heating characteristics than the 0.1 wt % MWCNT specimen. In addition, electrical resistance changes and magnetic field strength in 1.0 wt % MWCNT specimens can be measured because of the improved electrical properties due to increased network connectivity. Therefore, it can be concluded that MWCNT is thermally and electrically maximized when mixed with grout above an appropriate concentration.

## 4. Conclusions

In this research, temperature change analysis, electrical resistance change analysis, and magnetic field analysis were conducted to assess the filling rate of CNT cement grout. Based on the test results, the filling rate of the MWCNT cement grout was investigated with respect to each parameter. The following conclusions can be drawn from this study.

(1) The heating effect of the OPC grout and 0.1 wt % MWCNT specimens was not significant. Furthermore, it was impossible to evaluate the filling rate through thermal imaging analysis. Visual identification through thermal imaging is difficult because of the small temperature difference between the outdoor air and the ducts. Furthermore, void detection through electrical resistance was difficult because the resistance value was higher and close to the OPC specimen.

(2) The results of the heating experiment showed that the temperature increases in the 1.0 wt % MWCNT specimens were higher than those in the 0.1 wt % MWCNT specimens. This is because the number of CNT networks increased due to an increase in the MWCNT concentration.

(3) Thermal imaging analysis showed that the filling rate could be verified through visual inspection for the 1.0 wt % MWCNT specimens. An even temperature increase was observed in the 100%-filled and 75%-filled specimens, whereas a concentrated temperature change was observed close to the void in the middle of the 50%-filled and 25%-filled specimens. These results indicate that specimens with a concentrated temperature increase have low filling rates. Therefore, void detection should be possible using thermal imaging analysis.

(4) The electrical resistance change analysis and magnetic field analysis indicated that only 1.0 wt % of the MWCNT cement grout showed lower resistance and an obvious resistance change. Thus, more accurate estimations can be made when the filling rate is assessed through electrical resistance change analysis and magnetic field analysis.

(5) Cross-sectional analysis of the CNT cement grout using FE-SEM demonstrated the obvious heating effect of the 1.0 wt % MWCNT cement grout. The 1.0 wt % MWCNT cement grout had high thermal efficiency due to the network between the CNTs. These specimens also showed better electrical properties because of the increased number of network connections.

(6) The 1.0 wt % MWCNT cement grout is considered the optimum for grout filling evaluation because its heating effect is sufficient for assessing the filling rate with the naked eye using a thermal imager. Furthermore, the 1.0 wt % CNT concentration (which is a high concentration) should be used to detect voids in PC grout.

## Figures and Tables

**Figure 1 nanomaterials-10-00010-f001:**
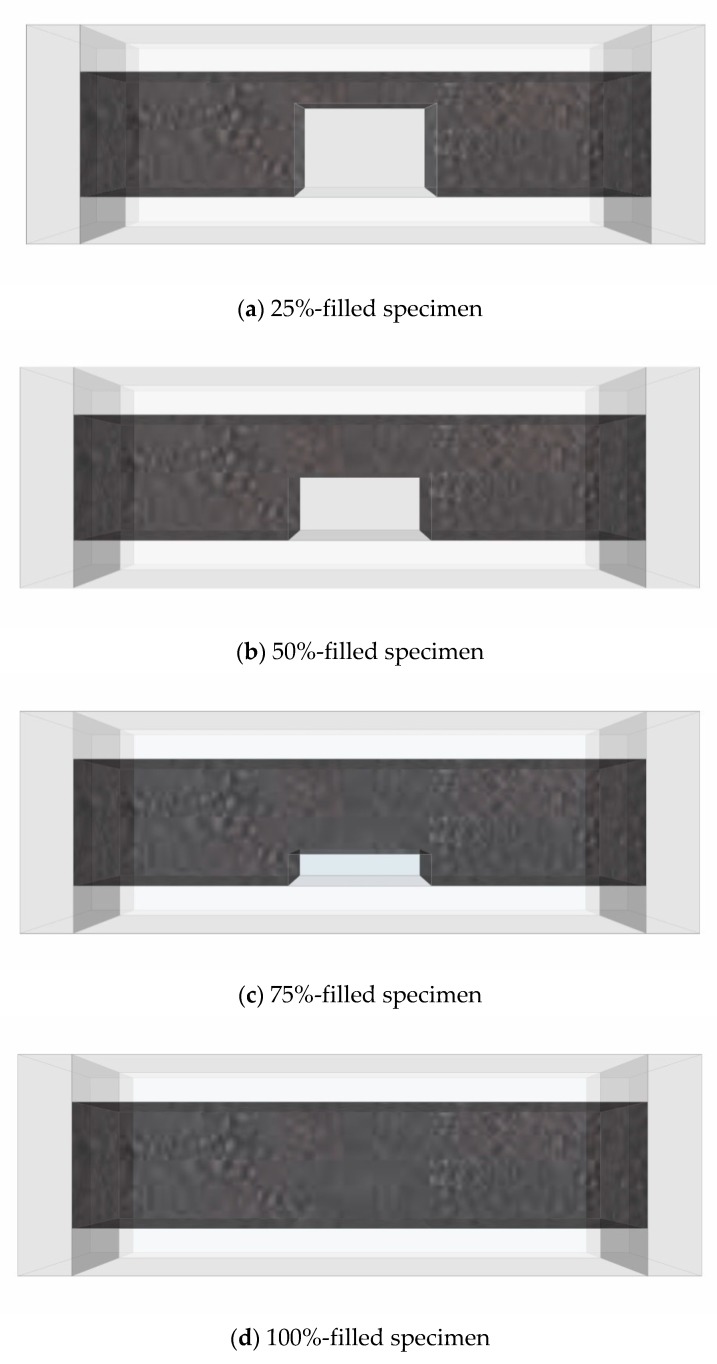
Void detection specimens of the Carbon Nanotube cement grout.

**Figure 2 nanomaterials-10-00010-f002:**
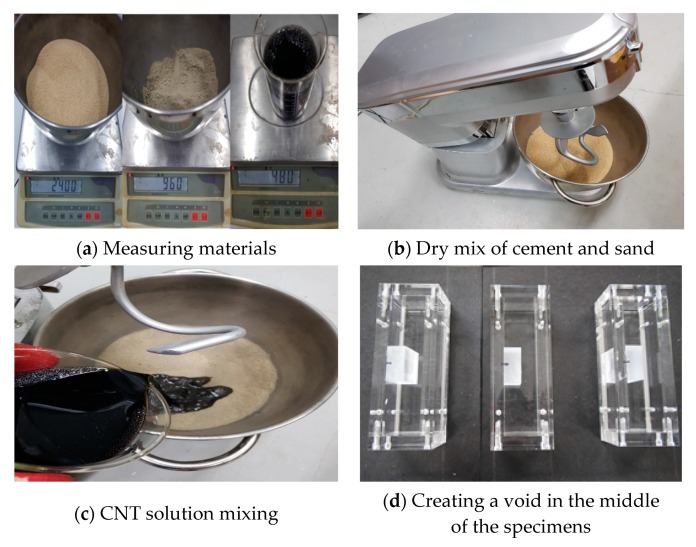
Specimen fabrication process.

**Figure 3 nanomaterials-10-00010-f003:**
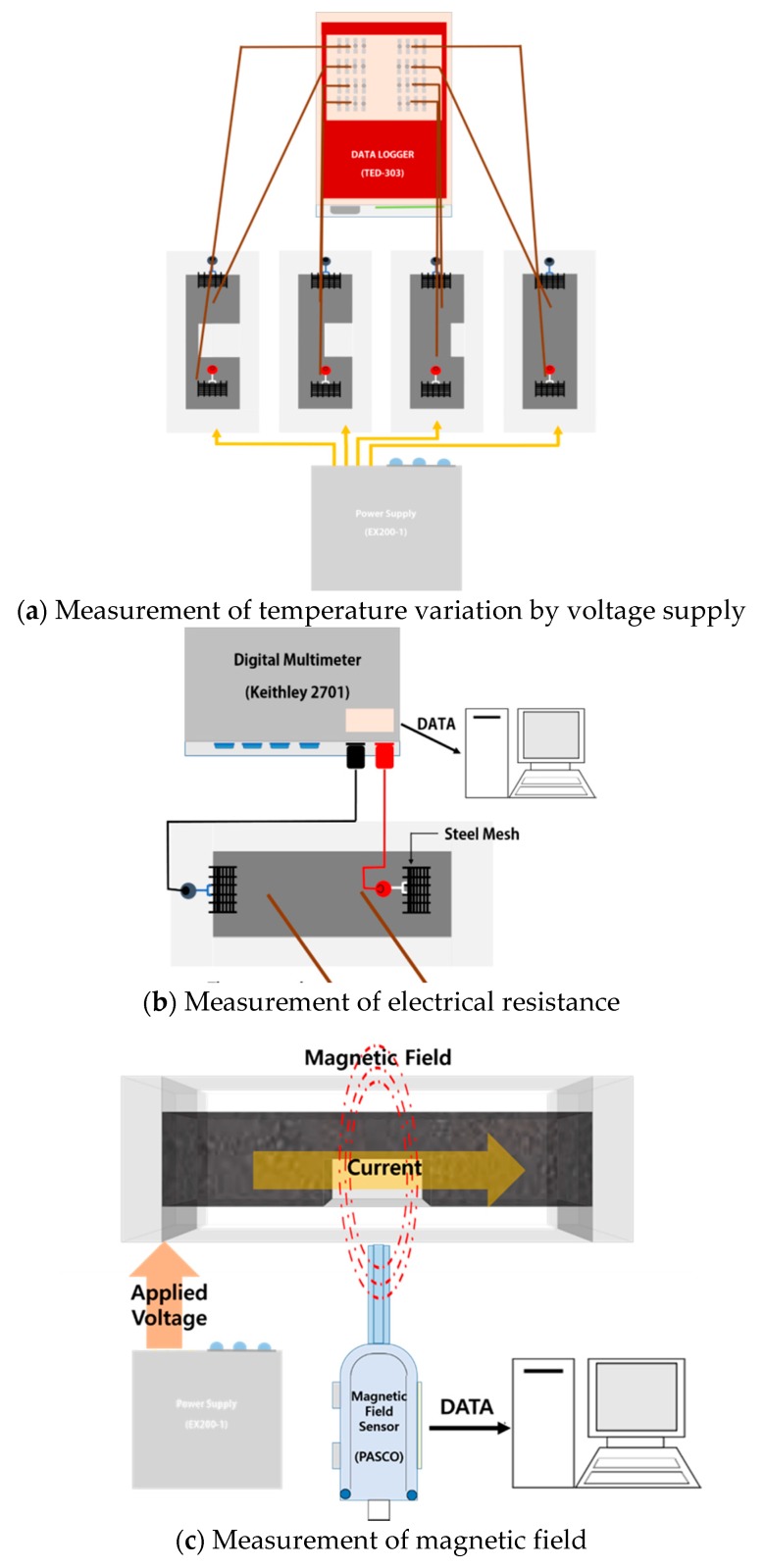
Test setup.

**Figure 4 nanomaterials-10-00010-f004:**
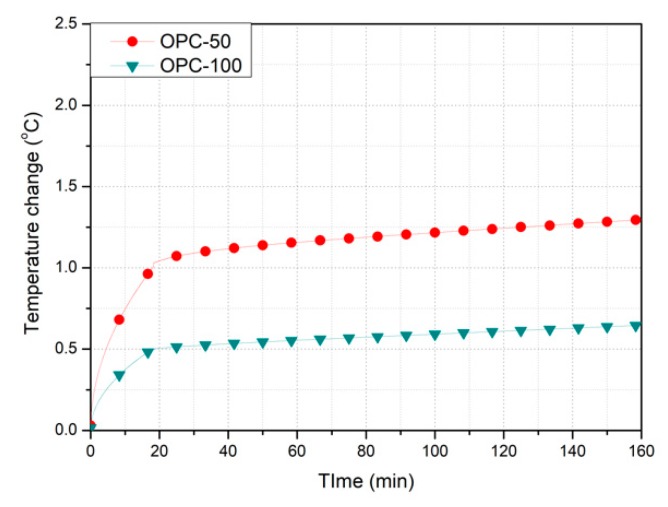
Temperature change of the Ordinary Portland Cement specimens.

**Figure 5 nanomaterials-10-00010-f005:**
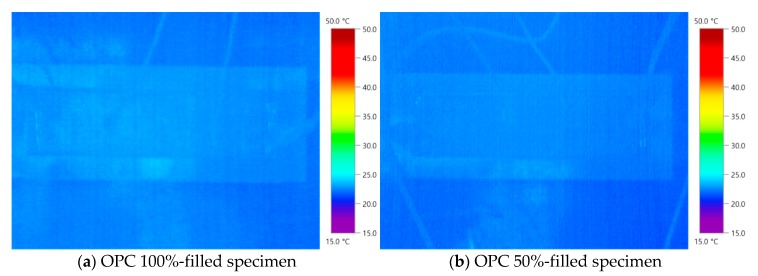
Thermal images of the OPC specimens.

**Figure 6 nanomaterials-10-00010-f006:**
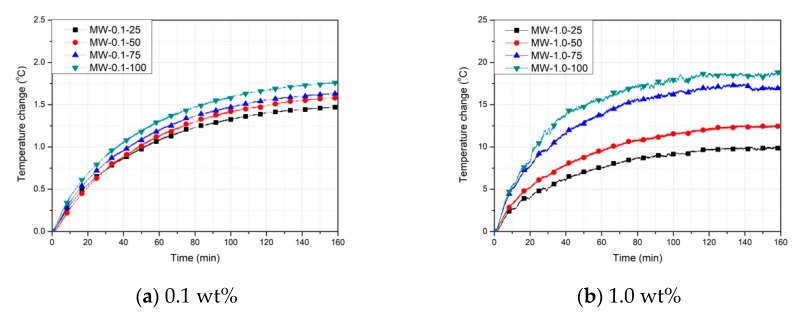
Temperature change of the MWCNT cement grout specimens.

**Figure 7 nanomaterials-10-00010-f007:**
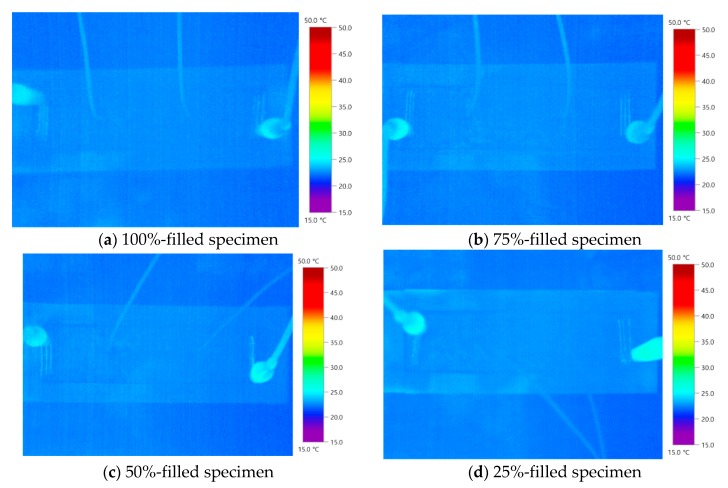
Thermal images of the MWCNT 0.1 wt % specimens.

**Figure 8 nanomaterials-10-00010-f008:**
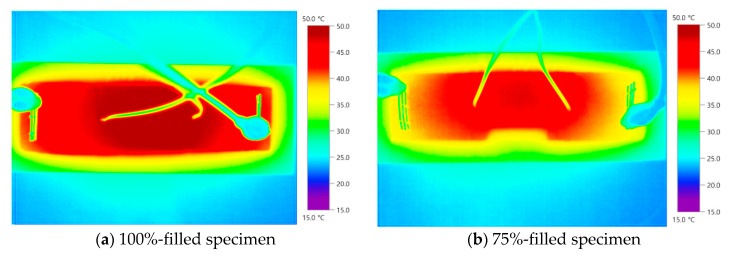
Thermal images of the MWCNT 1.0 wt % specimens.

**Figure 9 nanomaterials-10-00010-f009:**
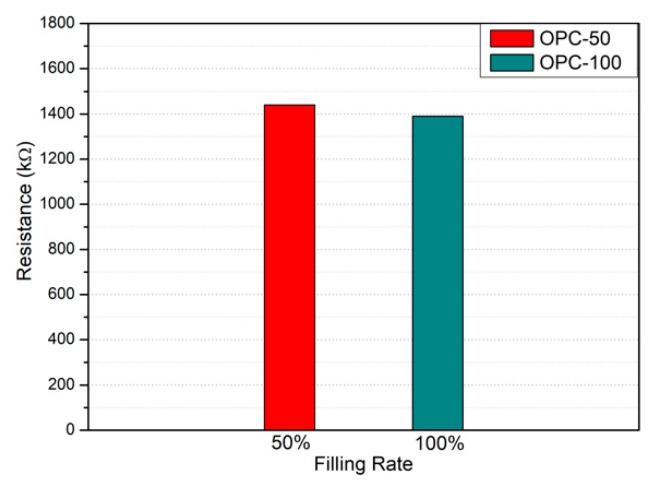
Electrical resistance of the OPC specimens with respect to the filling rate.

**Figure 10 nanomaterials-10-00010-f010:**
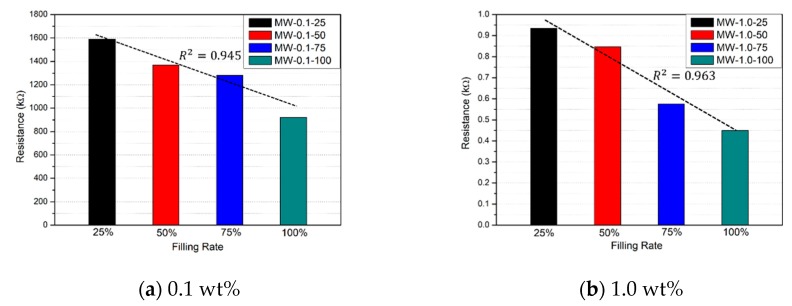
Electrical resistance of the MWCNT cement grout specimens with respect to the filling rate.

**Figure 11 nanomaterials-10-00010-f011:**
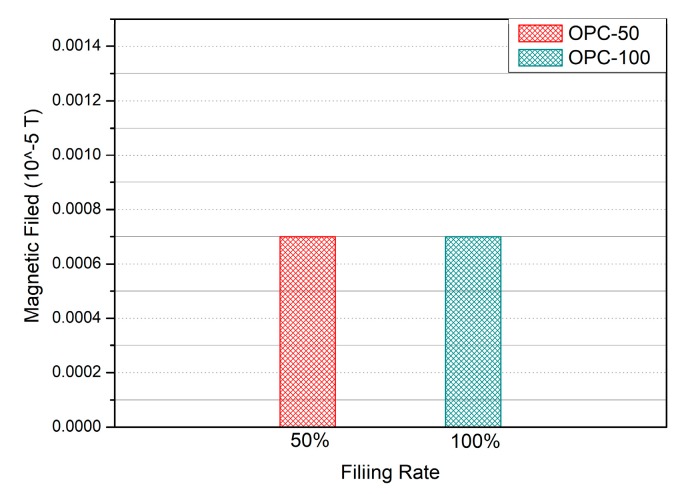
Magnetic field of the OPC specimens with respect to the filling rate.

**Figure 12 nanomaterials-10-00010-f012:**
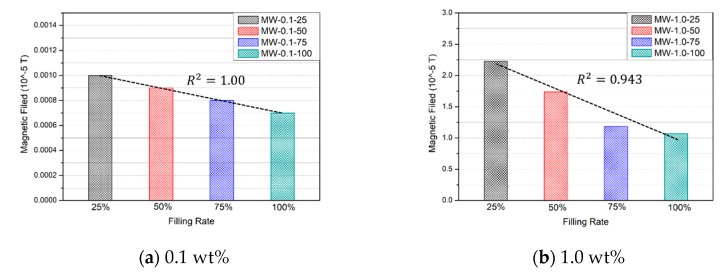
Magnetic field of the MWCNT cement grout specimens with respect to the filling rate.

**Figure 13 nanomaterials-10-00010-f013:**
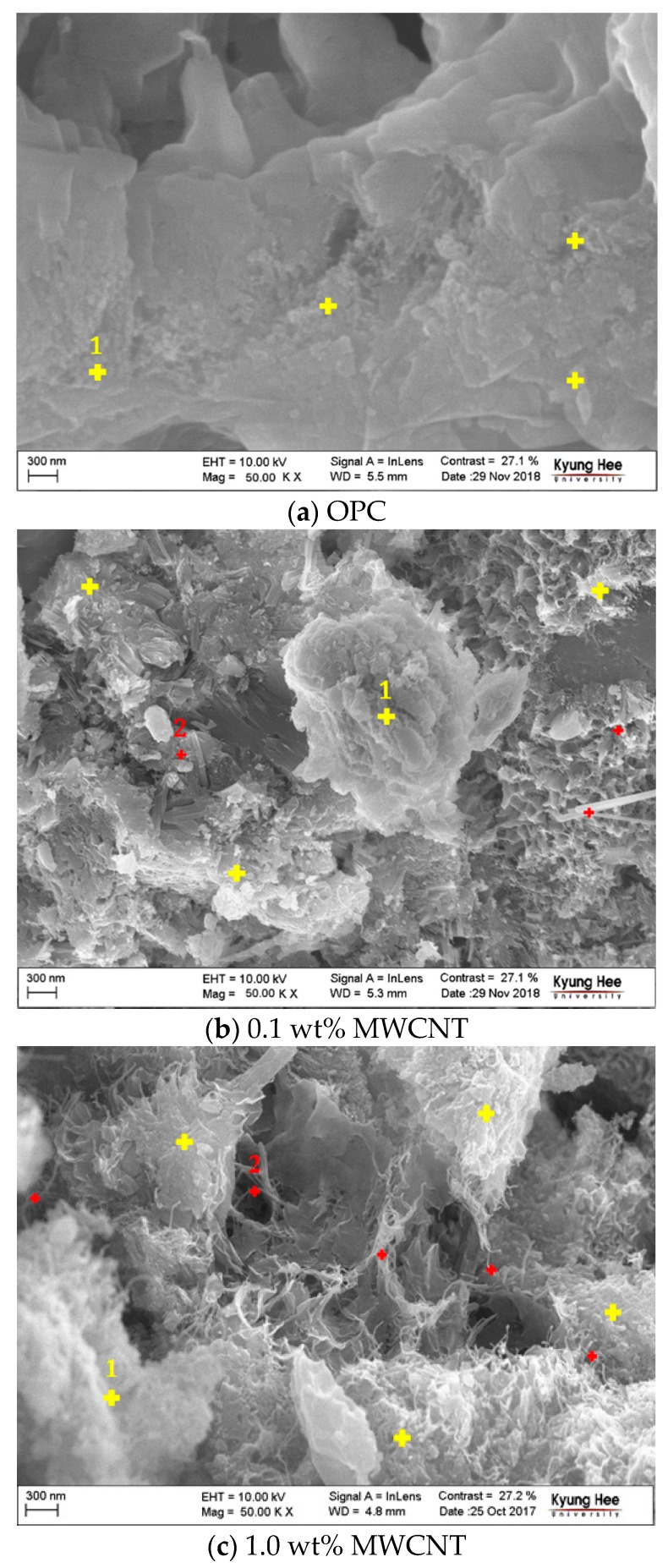
FE-SEM images of the specimens.

**Table 1 nanomaterials-10-00010-t001:** Physical properties of individual Multi-Walled Carbon Nanotubes.

	MWCNTs	Notes
Structure	Diameter: 5–100 nm Length: 100 nm–1 cm	-
Modulus of Elasticity	0.3–0.95 TPa	0.1–0.14 TPa (Cement mortar)
Tensile Strength	11–63 GPa	0.002–0.01 GPa (Cement mortar)
Electrical Conductivity	~6000 S/cm	5810 S/cm (Copper)
Heat Conductivity	Max. 3000 W/(m·k)	3320 W/(m·k) (Diamond)

**Table 2 nanomaterials-10-00010-t002:** Parameters of the Carbon Nanotube cement grout in void detection experiments.

Group	Specimen Name	CNT Type	CNT Content (wt %)	Filling Rate (%)
Group #1	OPC-100	-	-	100
	OPC-50			50
Group #2	MW-0.1-100	MWCNT	0.1	100
	MW-0.1-75			75
	MW-0.1-50			50
	MW-0.1-25			25
	MW-1.0-100		1.0	100
	MW-1.0-75			75
	MW-1.0-50			50
	MW-1.0-25			25

OPC: Ordinary Portland cement grout, MW: MWCNT cement grout.

**Table 3 nanomaterials-10-00010-t003:** Technical data of the thermal imager.

	Technical Data	Testo 882
Infrared image output	Detector type	320 × 240 pixel
	Thermal sensitivity	0.05 °C at + 30 °C
	Image refresh rate	9 Hz or 33 Hz
Measurement	Temperature range	−20–350 °C
	Accuracy	±2 °C, ±2% of the measured value

**Table 4 nanomaterials-10-00010-t004:** Thermal experiment results for the CNT cement grout with respect to filling rate.

Group	Specimen Name	Filling Rate (%)	Temperature Change (°C)	Standard Deviation	Specimen/100%-Filled Specimen
Group #1	OPC-100	100	0.65	0.002	-
	OPC-50	50	0.85	0.003	1.31
Group #2	MW-0.1-100	100	0.65	0.002	-
	MW-0.1-75	75	0.65	0.001	1.00
	MW-0.1-50	50	0.75	0.002	1.15
	MW-0.1-25	25	0.75	0.003	1.15
	MW-1.0-100	100	18.8	0.012	-
	MW-1.0-75	75	17.3	0.025	0.92
	MW-1.0-50	50	12.3	0.019	0.65
	MW-1.0-25	25	9.8	0.022	0.52

OPC: Ordinary Portland cement grout, MW: MWCNT cement grout.

**Table 5 nanomaterials-10-00010-t005:** Electrical resistance and magnetic field results for the CNT cement grout with respect to the filling rate.

Group	Specimen Name	Electrical Resistance (kΩ)	Magnetic Field $\left(10^{-5}T\right)$	Standard Deviation	Specimen/100%-Filled Specimen
Group #1	OPC-100	1390	0.0007	0.001	-
	OPC-50	1440	0.0007	0.000	1.00
Group #2	MW-0.1-100	1040	0.0010	0.000	-
	MW-0.1-75	1110	0.0009	0.000	0.94
	MW-0.1-50	1270	0.0008	0.001	0.82
	MW-0.1-25	1390	0.0007	0.001	0.75
	MW-1.0-100	0.449	2.2271	0.002	
	MW-1.0-75	0.575	1.7391	0.001	0.78
	MW-1.0-50	0.846	1.1820	0.001	0.53
	MW-1.0-25	0.934	1.0707	0.002	0.48

OPC: Ordinary Portland cement grout, MW: MWCNT cement grout.

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
