# Peer review of "Enhanced Detection Systems of Filling Rates Using Carbon Nanotube Cement Grout"

_nanomaterials, 2019, doi:10.3390/nano10010010_

Round 1

Reviewer 1 Report

Paper by Lee et al. concerns a scientifically straightforward, not to say obvious, but technologically important aspect of electrical/thermal/magnetic-based monitoring voids in cement grout with admixed multi-wall carbon nanotubes (MWCNTs). The technical importance of the paper is well-justified as the research concerns a key aspect of civil engineering, i.e. it might influence on the stability of the final architecture constructions. 

There are however few flaws and (sometimes significant) errors which should be addressed/corrected - otherwise the overall scientific value of the work would be minor and its visibility/'citability' rather poor, please find them below.

1) From l. 22 to l.47 there is no references at all(!). The gaps should be filled by key papers in the field of CNTs and its multi-functional composites.

2) l. 107: should be 'admixing MWCNTs' not 'mixing'.

3) Table 1 in its current form is pointless as it concerns general characteristics of MWCNTs. Moreover, the data for MWCNTs concern INDIVIDUAL MWNCNTs and not MWCNT powder/bulk form. But what is the most important, there is no characteristics of MWCNTs used in the paper (morphology, diameter/length, contaminotion, producer, etc.). No SEM/TEM data is on those MWCNTs.

4) The title is misleading since actually there is no filling rate monitoring - no real-time monitoring is presented in the paper, but only particular steps of filling the forms.

5) Figs 5 & 7 are badly prepared - the Y-axes should be up to 5 deg C and 5 (a) and 25 deg C (b), respectively.

6) I suggest putting the reference (blank) samples - with no change in characteristics - to Supplementary Data

7) Fig. 11 - why is the data presented as a bar-chart? By physics - you would expect a linear response with R2-correlation representing the error analysis.

8) No error analysis is presented for the measurements.

9) l. 391-2 This sentence is wrong - percolation threshold is the concentration level and this was not studied at all.

10) What happens with the mechanical performance of the final composite containing 1.0 wt.% MWCNTs? What about water penetration - MWCNTs are inherently hydrophobic and hence they might lead to poorer adhesion and larger non-filled in-cement spaces.

11) What about (even brief!) economical analysis of this, after all, industrial application?

Based on the above premises, I recommend a major revision for this work.

Author Response

The authors would like to sincerely thank the reviewer for the comments and suggestions, which contributed to a better quality paper. The comments of the reviewer have been fully incorporated into the revised version of the paper, as detailed below.

1) As recommended by the reviewer, the text and reference has been changed.

Manuscript location: First paragraph of Page 1, Reference

2) As mentioned by the reviewer, the text has been changed.

This study aims to fabricate MWCNT cement grout by admixing MWCNTs and to carry out void detection by conducting thermal imaging analysis, electrical resistance change analysis, and magnetic field strength analysis.

Manuscript location: Second paragraph of Page 3

3) The authors state the general characteristics of MWCNTs. The characteristics of the used MWCNTs in the experiments are also shown in Table 1. The SEM of the MWCNTs used is shown in Figure 13. Details of MWCNTs used in this study were added to the text.

Table 1 shows the mechanical and physical characteristics of MWCNTs [21,22]. The average diameter of the MWCNTs was 9.5 nm and the average length was 1.5 μm. The MWCNT samples were obtained from Dittotechnology Co., Ltd., and class 1 ordinary Portland cement was used for the cement.

Manuscript location: Forth paragraph of Page 3, Figure 13

4) In real sheath tubes it is possible to measure the filling rate of the grout using this technique. However, experiments were carried out with transparent tubes to prove the measurement technology with thermal imaging cameras

5) As recommended by the reviewer, Figure 4 and 6 have been modified.

Manuscript location: Figure 4,Figure 6

6) The reference sample is the filling rate of 100%. As recommended by the reviewer, Figure 4,6,9,10,11 and 12 have been improved. 

Manuscript location: Figure 4, Figure 6, Figure 9, Figure 10, Figure 11, Figure 12

7) The electrical resistance is kept constant depending on the material properties. Thus, the authors presented the results in a bar chart.As suggested by the reviewer, Figure 10 and 12 have been improved.

Manuscript location: Figure 10, Figure 12

8) Nano-concrete shows clear results compared to reference. Therefore, it is not shown because the error on the measurement is very small.

9) As recommended by the reviewer, the sentence has been removed.

Manuscript location: Conclusions 2

10) The purpose of this paper is to measure the filling rate of grout. Therefore, a paper on the strength characteristics of the same MWCNT cement used in this experiment is indicated as a reference.

Manuscript location: Reference 48, Last paragraph of Page 7

11) MWCNTs are relatively expensive than ordinary construction materials. However, CNT prices are gradually falling, and MWCNT is currently available for industrial application.

Reviewer 2 Report

Dear Prof. Johnathon Shan,

I have reviewed the article entitled “Enhanced Detection Systems of Filling Rates using Carbon Nanotubes Cement Grout" by heeyoung lee, Seonghoon Park, Sanggyu Park, and Wonseok Chung (Manuscript ID nanomaterials-656370).

Here my comments:

In general, the manuscript is well written. The introduction part is too short, without showing the relevancy of the research and explanation of the materials selection. It is recommended to add more relevant manuscripts.

Page 7, lines 177-181: Please elaborate which surfactant was used for CNT dispersion and its concentration. Previous work showed that long duration chops down the CNT’s, thus 4 hours sounds too long.

It is recommended using vibrating table during the cast of reference samples, since it reduces the voids.

Page 9 line 203: the 4-point technique is used mainly for surface resistivity measurements. Thus, it can affect the accuracy of measurements. It is better using volume resistivity apparatus.

Page 12, table 4: The temperature changes are missing the stand deviation. Please add them to the table.

Page 16, table 5: Please fix the table’s borders. Also, please add the standard deviation values.

Page 18, Figure 14: It is recommended adding reference image of reference sample and 0.1% CNT.

The low filling rate of 1% CNT can be attributed to increased viscosity. There is no reference in the text regarding the rheological properties.

The manuscript lacks the mechanical properties of the specimens. It is recommended adding flexural properties W/O the changes in conductivity and/or magnetic field properties, simultaneously.

To asses the real content of voids, one can use density measurements. Please complete if possible.

There are too many figures. Please consider removing some (such as Figure 2, 5, 6 and 10).

The temperature axes range in Figure 7 can be reduced.

Major revision is needed before resubmitting the manuscript.

Author Response

The authors would like to sincerely thank the reviewer for the comments and suggestions, which contributed to a better quality paper. The comments of the reviewer have been fully incorporated into the revised version of the paper, as detailed below.

1) As recommended by the reviewer, introduction part has been modified.

Manuscript location: First paragraph of Page 1, Reference

2) As recommended by the reviewer, some sentence have been modified.

The homogenous dispersion of nanomaterials is very important for cement composites mixed with MWCNT concentration of 1%, which is high concentration. For this purpose, surfactant and ultrasonic dispersion energy were selected through numerous experiments.

The surfactant used in this study is the sodium dodecyl sulfate (SDS). Although the four-hour ultrasonic wave dispersion seems excessive, it is considered appropriate for the distribution of homogenized MWCNT inside the cement structure.

Manuscript location: Second paragraph Page 7

3) The 4-point technique is the most reliable method of measuring electrical resistance, and most cement resistance tests are used as a 4-point technique. As recommended by the reviewer, authors will compare 4-point technique with the volume resistivity apparatus in a later study.

Manuscript location: First paragraph of Page 9

4) As recommended by the reviewer, the stand deviation has been added.

Manuscript location: Table 4

5) As suggested by the reviewer, the stand deviation has been added.

Manuscript location: Table 5

6) As recommended by the reviewer, Figure 13 has been modified.

Manuscript location: Figure 13

7) The purpose of this paper is to measure the filling rate of grout. Therefore, a paper on the strength characteristics of the same MWCNT cement used in this experiment is indicated as a reference.

Manuscript location: Reference 48, Last paragraph of Page 7

8) The study of the mechanical properties of cement composites using MWCNTs has been fully cited in the references. This paper mainly aims at measuring the filling rate of grout.

Manuscript location: Reference 1-10, 48, Last paragraph of Page 7

9) The purpose of this paper is to measure the filling rate of grout. Therefore, a paper on the density measurements of the same MWCNT cement used in this experiment is indicated as a reference.

Manuscript location: Reference 1-10, 48

10) 

As advised by the reviewer, Figure has been removed

As recommended by the reviewer, Figure 6 has been modified.

Manuscript location: Figure 6

Reviewer 3 Report

This paper reports the effectiveness of CNT cement grout to detect the void and the filling rate. The idea is very interesting and of interest to the CNT community. The results are clearly presented and the conclusions are sound. Therefore, I would recommend it for acceptance after addressing following points.

The authors should show the company of the MWCNTs used here because the physical properties of MWCNTs strongly affect the experimental results. If they synthesized CNTs themselves, the details of the synthesis methods should be shown. The authors should carefully check Table 1. I think 5,810 S/cm is too small for the electrical conductivity of Cu. I cannot understand why the authors concluded that void detection was difficult for 0.1 wt% MWCNT specimens (p.15). The electrical resistance shows clear filling rate dependence (Fig. 11a). The caption of Fig. 13 may be wrong.

Author Response

The authors would like to sincerely thank the reviewer for the comments and suggestions, which contributed to a better quality paper. The comments of the reviewer have been fully incorporated into the revised version of the paper, as detailed below.

As recommended by the reviewer, some sentence have been added.

The MWCNT samples were obtained from Dittotechnology Co., Ltd., and class 1 ordinary Portland cement was used for the cement.

Manuscript loacation: Last paragraph of Page 3

The properties of Synthesized CNTs have already been mentioned in many papers. Therefore, this paper is cited by reference. As recommended by the reviewer, Table 1 was carefully reviewed.

Manuscript loacation: Reference 1-10, 48

0.1 wt% MWCNT specimens may be capable of void detection. However, as shown in Table 5, 1.0wt% is relatively better than 0.1wt% MWCNT specimen. Therefore, the conclusion of the study was to determine 1.0wt% as the optimal concentration.

Manuscript loacation: Conclusion 2,3,4

As recommended by the reviewer, Figure 13 has been modified.

Manuscript loacation: Figure 13

Round 2

Reviewer 1 Report

The authors MUST address the points which were neglected in the revised version of the manuscript.

Table 1 concerns data from INDIVIDUAL MWCNTs and it should be emphasized in the title of the table

Fig. 4 & Fig. 6 were not corrected and hence the differences between samples cannot be visible. The Y-axes should be of a range 0-2.5 deg C for both figures.

Author Response

Q1. The authors MUST address the points which were neglected in the revised version of the manuscript.

A1. The authors would like to sincerely thank the reviewer for the comments and suggestions, which contributed to a better quality paper. The comments of the reviewer have been fully incorporated into the revised version of the paper, as detailed below.

Q2. Table 1 concerns data from INDIVIDUAL MWCNTs and it should be emphasized in the title of the table.

A2. As recommended by the reviewer, Table1 and text changed.

Table 1 shows physical properties of individual MWCNTs [21,22]. Location : Table 1, Fourth paragraph of Page 3

Q3. Fig. 4 & Fig. 6 were not corrected and hence the differences between samples cannot be visible. The Y-axes should be of a range 0-2.5 deg C for both figures.

A3. As mentioned by the reviewer, Figure 4 and Figure 6 has been modified.

 Location : Figure 4 of Page 12, Figure 6 of Page 13
